# The Scientist: Creator and Destroyer—"Scientists' Warning to Humanity" Is a Wake-Up Call for Researchers

**Marek Cuhra** 

Marbank, Institute of Marine Research, 5005 Bergen, Norway; marek.cuhra@gmail.com

**Abstract:** Scientists investigate, describe, invent and create. Most advances in medicine, technology and understanding of the living world in the context of the cosmos, are attributable to systematic efforts by expert researchers. However, pervasive toxins, persistent environmental pollution, destructive weaponry and resource depletion are also outcomes of scientific efforts. Furthermore, although we have reached great advances in some research fields, other issues are enigmatic and arguably could be investigated with other methods or mindsets. That, however, brings us to a paradoxical realization: Despite the fact that there are more scientists in this world than ever before, due to socialization and indoctrination we are currently suffering from reduced cognitive diversity within academic disciplines. Arguably, scientists are not taught to think independently and differently, instead we are educated into a compliant, univocal and homogenous, 'Wissenschaftlicher Denkkollektiv.'

**Keywords:** scientific skepticism; interdisciplinary research; agnogenesis in academia; moral responsibility of scientist

## 1. Introduction

The current commentary is written by a scientist with three decades of experience from ecology, genetics, toxicology, pharmacology, waste-management, consultancy, scientific diving and risk-assessment research in private and public sector institutions. The commentary will briefly touch upon six complex issues which arguably all by themselves deserve deeper scrutiny: In Section 2 on scientists as professionals as well as ordinary mortal beings, in Section 3 on scientific methods as investigation as well as avoidance, and in Section 4 on quantification versus rationality (which is further expanded on in Section 5, which deals with educative cultures). Section 6 highlights the importance of correctly interpreting the perceived subject (be it through observation or through analysis of empirical data). Section 7 mentions aspects of the scientific workflow (theory, hypothesis development and experimental work), and speculates whether these three stages of scientific work necessarily need be bundled into a single 'research outcome,' such as the framework of a traditional scientific paper. The concluding remarks further attempt placing the current commentary in relation to the recent "Scientists' Warning to Humanity"—thus this commentary will start with the "World Scientists' Warning to Humanity" [1] and continue as a scientific dive into local subjects:

In a recent brief assessment of the state of the Earth and the natural environment vitally important for human life, fifteen thousand signatory scientists have concluded that if the course of present developments such as environmental resource depletion, accelerating consumerism and continued habitat destruction is not drastically altered, the planet will no longer be able to supply us with vital ecosystem services and consequently humanity will *perish* [1].

There are several noteworthy aspects of this disturbing development, not least the fact that a primary motivation for the destruction which humanity is inflicting on its own living environment,

arguably is a simple strategy for financial gain: Dominant industrial activities, agroindustry, petrochemical industry, consumerism and extraction of resources invoke tremendous strain on ecosystems and disrupt the services which nature performs [2].

Thus, mankind destroys Earth for *profit*. Paradoxically, we, the scientists, are fundamentally instrumental in inventing, developing and expanding the industrial materialism which so obviously is cutting the branch on which humanity is comfortably seated [3]. Consequently, analysis of scientific cultures is persistently relevant, and this year, as any year, academia should highlight the crowning achievements of the great proponents of scientific method and research ethics: In 1969, half a century ago, professor Karl Popper (1902–1994) published his essay on the *Moral Responsibility of the Scientist* [4], following his 1962 essay on *the Sources of Knowledge and Ignorance* [5]. Those reflections form fundamental broad roots for further developments in philosophy and epistemology and are as relevant today, as they were fifty years back in time.

Importantly, Popper argues that we, the scientists, must *care* and involve ourselves in our work: It is perfectly feasible to remain objectively truth seeking, as any academic professional must, while also advocating honesty and further moral objectives. Obviously, such personal involvement in our specialized subject should not preclude assistance for regulatory actions: Undoubtedly, we, the scientists, comprehend the issues that we are investigating better than most other mortals. And, how can politicians or the public be expected to act according to scientific findings, if we do not elucidate results and consequences of those findings?

In 1969 Popper defined *war* as the main threat against human existence and warned against nuclear conflict. In the contemporary world of 2019, humanity is arguably still on the verge of nuclear war, although we must contemplate that nuclear weapons are technology which is almost eight decades old: How old is the technology of our cars, televisions or telephones in comparison? Given the fact that no singular sector in society—neither medical research, environmental protection, genetics, biochemistry or any other specific discipline—receives as much funding as the clandestine research which develops more efficient technology intended to destroy and kill, we must acknowledge the high probability that infinitely more devious weapons technologies have been developed in recent decades—weapons which are kept in utmost secrecy and may harbor destructive forces far beyond our imagination.

Furthermore, a glittering collection of new threats has arrived onto the global arena in the five decades that have passed since Popper drew his conclusions. Today, genetic manipulation, climate manipulation, health manipulation and a complex intoxication of the total environment, add risk to our lives. Only expert scientists may sufficiently understand such specific issues, and thus we—the scientists—arguably have a moral obligation to assist society in gaining the necessary understanding and to enable our politicians to make optimal decisions.

Unfortunately, it seems that, at the moment, only a few scientists actively engage in dialogue with wider society. Also, it must be argued that natural science in general is not conserving or restoring nature, rather it is more-or-less willingly contributing to increased destruction. To understand this paradox, we must understand what science *is*, how scientists are *produced*, how we are *indoctrinated* into a utilitarian view of nature and how we have become restricted and corrupted by a general lack of oversight and genuine understanding.

In contemporary societies, a researcher's position is often merely an occupation in line with much office employment: Although some scientists are deeply engaged in their occupation and get completely absorbed in it, still, most individuals employed as researchers in natural sciences lead ordinary lives. Arguably, a majority of us thus work comfortably nine to five, with a modestly low ambition of developing our own little square of home-turf expertise, and few of us demonstrate holistic reflections or effort in ensuring that the consequences of our actions lead to true benefits for society.

## 2. The Scientist

*"It would be folly to argue that our knowledge is sufficient to allow any expert, in any realm of social importance, to claim finality for his outlook. He too often, also, fails to see his results in their proper*

*perspective [ ... ] The expert, in fact, simply by reason of his immersion in a routine, tends to lack flexibility of mind once he approaches the margin of his special theme."*

—Harold J. Laski, The Limitations of the Expert, 1931.

Scientists are expected to investigate, describe and explain, and thus produce new knowledge. Facing even the most difficult of intellectual challenges we systematically focus all of our attention, we submerge in near-unfathomable detail and we isolate ourselves in the complex laboratories of our profession. We scrutinize the deepest oceans, the most remote corners of the Universe and the microscopic details of cells, along with materials, phenomena and most aspects of the natural world. We invent and synthesize technology, chemicals and pharmaceuticals.

We work in focused and systematic procedures, collect data, draw charts and present our findings in meticulously deliberated papers. And, having successfully solved one of the unfathomable riddles of the Universe, we triumphantly emerge with new knowledge as well as inventions which will potentially benefit humanity and make this world a better place. Ideally, everything we do is intended to expand the knowledge of society and give a deeper understanding of specifics, while contributing to the heritage that our predecessors have left behind.

Subsequently, based on such scientific findings, bureaucrats and administration will formulate evaluations and advice for politicians to make informed decisions. It should be unnecessary to state that it is imperative that we are precise, truthful and trustworthy in everything we do, write and say. We have all been squeezed through the educational machinery and obtained degrees which allow entry to academic communities, attractive employment and vast possibilities.

Thus, we have all studied the work of peers and predecessors in order to gain understanding and inspiration. However, we must also be able to navigate independently in many questions pertaining to fundamental assumptions or bordering onto the ethics and morality of our profession: We must not blindly follow the advice of predecessors or colleagues; we must make up our own informed mind, or at least heed the minds of the thinkers who are most convincing and make most sense.

Worryingly, it has become obvious that some fellow scientists amongst us are not as rigorous, engaged and investigative as they arguably should be: Uncountable numbers of scientists misbehave, falsify and mislead [6]. Some colleagues will avoid issues which may be perceived as overly challenging and instead keep their attention comfortably within those phenomena and subjects which they master and fully comprehend. Also, some scientists withhold important evidence if it is found to be contradictory to interests which they represent—typically commercial interests or possibility further funding [7]. Further, it has unfortunately been demonstrated that many published findings cannot be reproduced in subsequent studies [8].

These ethical challenges and difficult matters are not to be ignored, as they obviously bear the potential to lethally damage our profession and general credibility. Possibly, most of us have faced situations in which we had to make subjective decisions in the borderlands of research ethics: Whether to include data from an experiment which didn't go exactly as planned, or whether to refer to certain work from competitors. Handling such small, everyday dilemma in certain ways might develop into a professional culture which is honed to produce the results which we find useful, but which may alienate us from objective search for knowledge and thus compromise our professional and personal integrity.

In his 2005 book on climate change, Tim Flannery claims that "scientists are in fact trained skeptics, and this eternal questioning of their own and others' work may give the impression that you can always find an expert who will champion any conceivable view" [9]. Although we might agree that scientific views are divided on numerous issues, it can be argued that Flannery is overly optimistic on behalf of his fellow scientists when it comes to independent thinking and healthy skepticism towards dogma. Unfortunately, far from all scientists conduct themselves in their professions according to the ideal of the trained skeptic defined in; e.g., The Norms for Researcher Conduct, compiled by Robert K. Merton (1910–2003) [10]. It could even be claimed that a dominant proportion of researchers are merely 'following the herd' in the mainstream of science.

Although skepticism is so very important to any scientific progress we must ask, *can* average scientists truly act as trained skeptics? To answer such a question, you must investigate what we scientists really are. If you look at us as a population, you might discover that we are terrifyingly uniform: In university we all learn the same assumptions and read the same books, written by authoritative voices who in their turn have all read the same books and all been taught the same fundamental truths. How could we even hope for truly contrasting views arising within such homogenized scientific communities?

Arguably, scientists in general are not behaving as trained skeptics, but more like a flock of geese. Once the flock is airborne and moving, it will keep its focus and direction. The individual geese in the flock might take turns in leading the way and breaking the turbulence, but the main direction is set and will not be changed unless something extraordinary comes up. The result is obviously counter-productive and fundamentally anti-scientific.

This brings us back to a fundamental observation on the genesis of scientific facts and the phenomenon of the "Wissenschaftlicher Denkkollektiv," as defined in 1935 by Ludwik Fleck (1896–1961). Fleck wrote on epistemological issues, such as scientific method, and on the socialization of the mind which enters the researcher community [11]. Arguably, such socialization prunes the creative free intellect as an unfortunately steep entrance fee into many academic workplaces. Probably, this writer is not the only scientist who has experienced rejection on basis of argumentation, such as, "How can we hire such a person as the postdoc, when he has a completely different background than we?" Implicitly—"In our department we do not recruit different minds; we recruit minds who think like us." Arguably, such approach guarantees cognitive stagnation through poorly premeditated elimination of institutional intellectual diversity.

Thus, if you lead a research group, you are not fulfilling your mandate unless you have a strong focus on recruiting minds that are as sharp as possible, and who know things which you yourself do not yet comprehend (Figure 1). Alternatively, if you recruit from your own environment and amongst friends in the backyard, then you probably will reduce the risk of innovation and progress. Additionally, you will also have less need to update yourself on new and possibly difficult advances within your field of specialization—and, you can use the same slides for your courses that you were using last year, and the year before that.

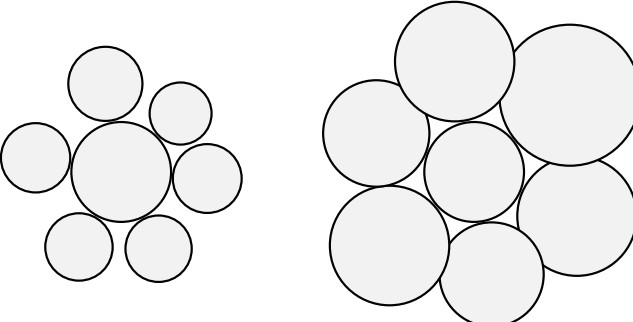

**Figure 1.** A main cause for intellectual dormancy in academia: Although of identical size—due to the surroundings, the central circle in the *left* figure seems of larger diameter than the central circle in the *right* figure. This optical illusion illustrates the academic delusion of inflated ego (accomplished in this example by recruiting colleagues who are less clever than one self).

## 3. The Search for Truth

*"Formerly, the pure scientist or the pure scholar had only one responsibility beyond those which everybody has; that is, to search for truth."*

　　　　　　　　　　　　　　　　　　　　—Karl Popper, The Moral Responsibility of the Scientist, 1969.

Somewhere in history it came to be, that we began to use empirically based scientific methods to understand and explain the wonders of nature: Natural science as we know it, is an invention created by the western mind. In contemporary sciences, practitioners have accepted the dualistic view of the world surrounding us, separating the material realities from philosophy and theory of metaphysical matters. And, depending on subjective opinion, either blame or credit for the present situation may be bestowed upon our predecessors in science: There is some confusion in academic circles regarding the origin of the present division, which has variously been attributed to guidance gleaned from the writing of predecessors such as René Descartes (1596–1650) and even Alexander von Humboldt (1769–1859).

Although alternative ways of understanding nature are still found amongst other cultures, in people who believe plants and animals to be spirited and divine powers helpfully contributing, still, most people living on this planet at present adhere to the culture which we—for lack of a better word—must term *consumerism*; a culture in which most of us seemingly thrive and actively contribute to the developments which, as initially said, have been described by the global scientists initiative as fundamentally destructive and contradictory to continued life on planet Earth [1].

Science as a mosaic of disciplines evolves, differentiates, widens and continuously accumulates knowledge. If we were to depict that knowledge according to principles of visualization of scientific information, we might draw a simple circle as a graphic representation of a sphere. The sphere consists of an outer layer, on which scientific disciplines such as genetics, quantum physics, astronomy, protein chemistry, invertebrate zoology and toxicology float as adjacent continents on a planetary surface, along with nuclear physics, endocrinology, agroecology and thousands of more or less distinct fields of scientific research. Those disciplines are constantly evolving and amassing published evidence, thus expanding the sphere by volume and area.

If we magnify a randomly chosen part of the sphere, as in Figure 2, we may notice black and white blemishes scattered on the surface; these are the commonly overlooked and ignored areas weakened by white lacunae and dark matter aggregations. To understand the nature and origin of these important and immensely destructive phenomena, we must perform an epistemological dissection and penetrate below the surface layers of contemporary scientific knowledge. Inside the sphere, in the deeper layers of science as it was in 2004, 1936, 1904, 1859 or 1756, we find evidence of that specific time and age.

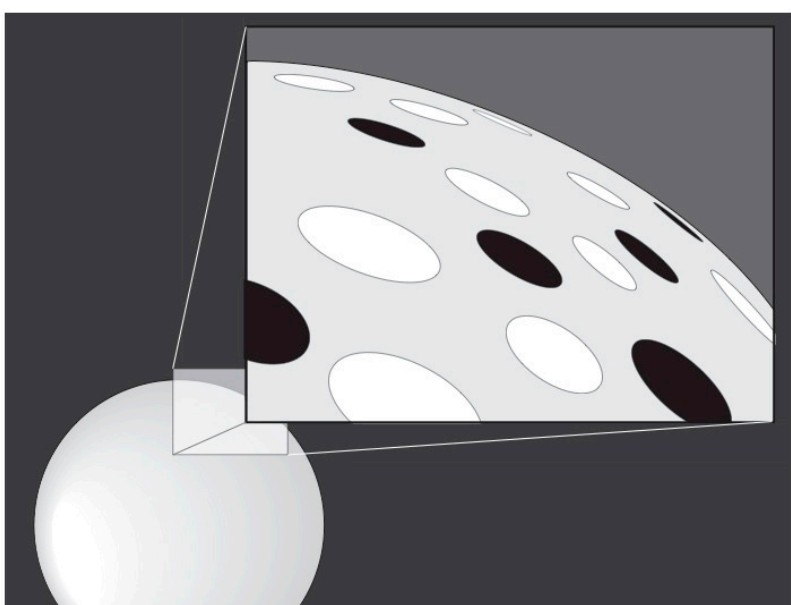

**Figure 2.** An external view of the expanding multi-layered structure of academic knowledge in natural sciences.

As paleontologists or archaeologists, we may peel away those outer layers and dig our way back into the prehistory which is shaping our present profession, to read again the old scriptures from predecessors who published 35, 84, 153 or 247 years ago. Or, instead of peeling away individual layers, we could mentally slice the sphere of scientific knowledge in a sort of epistemological dendrochronology.

Evidently, when we regard these old layers of scientific knowledge from previous centuries, we immediately notice that in those earlier times, the total volume of knowledges was smaller, much smaller. Arguably, in those earlier days of science, practitioners of sciences had less *endemic knowledge* of their own specialized subject to deal with and could thus afford to spend time studying *adjacent knowledge* and thereby gain and uphold a certain level of interdisciplinarity—a 20th century botanist would know a bit about not only entomology, soil chemistry and hydrography, but possibly even some marine invertebrate taxonomy as well.

Today, well into the 21st century, such professional diversity is seldom found: The contemporary specialist researcher rarely has a broad perspective nor much more than superficial knowledge of disciplines not strictly connected to his or her profession.

The fragmentation and specialization of sciences is not the main challenge stemming from the model of the expanding sphere—there are two other aspects which are more challenging for our professions—that of the *dark matter aggregations* and the *white lacunae* (Figure 3).

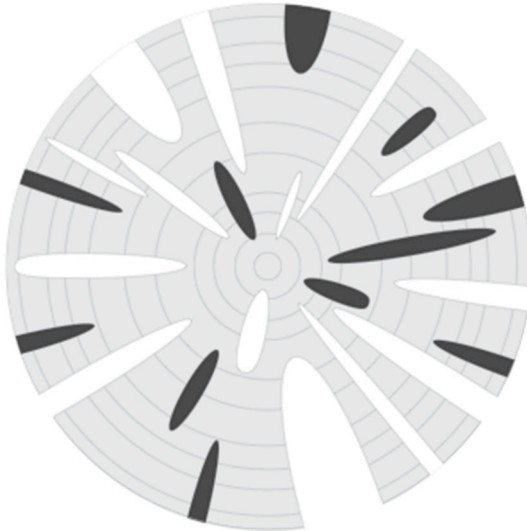

**Figure 3.** Slicing the sphere of Figure 2 we see that embedded white lacunae and dark-matter aggregations infest, confuse and invalidate part of our academic knowledge.

The lacunae are easily understood. They are blank vacuoles, air-filled cavities within the sphere of knowledges—the unexplored territories; the issues and scientific fields which we have avoided or overlooked or simply had to pass due to a lack of either understanding; e.g., [12], or which we have avoided due to lack of methods for investigation, such as the enigmatic plasma vortices described by Levengood and Talbot in 1999 [13]. Alternatively, they arise from scientific questions preliminarily approached in theory, and only much later investigated with modern methods; e.g., the theory of panspermia originally presented in 1908 by Arrhenius [14] and revived a century later [15,16].

Although standing out as obstructive voids and pieces missing from the big puzzle, the lacunae are not as serious a defect to our scientific knowledges as the dark matter aggregations. The white lacunae are missing areas of which we have no knowledge, but at least we are aware of this fact. Contrary to this, the dark-matter aggregations consist of facts and knowledge which scientists have produced and published, but which is erroneous and false. Thus, the dark-matter aggregations are hidden in the scientific heritage, disguised as scientific facts, and only the most investigative skeptics

amongst us can root them out: The dark-matter aggregations are lumps of anti-science, a sort of cancerous metastasis blemishing the credibility of academia.

These dark matter aggregations have arisen from erroneous understanding, falsified data, misinterpretations, arrogance or plain incompetence. Consequently, these blemishes are manifestations of *agnogenesis*—the production of ignorance [17]. Worryingly, some of the dark matter aggregations are situated in scientific domains which are vitally important for society, such as those pertaining to health, nutrition and environmental protection.

## 4. Quantification and Rationality

*"The treasure of empirical contemplation, collected through ages, is in no danger of experiencing any hostile agency from philosophy."*

—Alexander von Humboldt, Cosmos, 1845.

Fear of theory is not a general problem in academia, but in natural sciences it seems that only few scientists dare to present writing based on less tangible evidence than data amassed through experimental testing. Data is a curious contrivance and arguably, the mere idea of quantification as an exclusive strategy for understanding nature should be thought provoking, as noticed more than a century ago by St. George Mivart [18]. Never the less, at present a dominant scientific tool for the study of natural phenomena, is mathematics.

Although evident for most academics, the key role of mathematics in management and mismanagement of this planet must still be discussed: Arguably, the main defect of mathematics is the belief upheld in many academic circles, that natural sciences based in mathematics are the only reliable means towards answering the big questions, such as *why* and *how*.

It may appear offensive and inappropriate to argue against such thinking by claiming that mathematics should mainly be used for measurements, comparisons, statistical probabilities and other such simple duties. Accordingly, mathematics is well suited for mundane tasks, such as counting polar bears and concluding that only half are left. After meticulously reporting and communicating such a fact, the mathematician-biologist again sits at his desk for some years, and then makes another survey counting polar bears. He then finds, that there are only half as many as when he previously counted them, but that, surely, can be interpreted to signify that the population is stable around a curve correlated with coffee breaks and international meetings. Provided half the bears are constantly left, the advice is still that seals should be culled, to reduce their consumption of fish which could be commercially harvested. For those suspecting this interpretation as merely morbid sarcasm, the calculations and conclusions presented in numerous papers by fisheries' statisticians can serve as a sextant measure of scientific decline.

As an alternative to employing mathematicians to serve as biologists, the entire education system in natural sciences should be fundamentally evaluated. The biologists who give advice on the management of resources must primarily *understand and respect nature* which they are mandated to manage, and not necessarily excel with *numbers*. However, numbers are easy to work with and in a complex world they produce simple answers. Furthermore, for some academics as well as politicians, numbers are more tangible and approachable than the hazy clouds obscuring the aforementioned big questions of *how* and *why*.

## 5. Scientific Cultures

*"The British school insisted that the ultimate source of all knowledge was observation, while the Continental school insisted that it was the intellectual intuition of clear and distinct ideas."*

—Popper, the Sources of Knowledge and Ignorance, 1962.

Although Popper propagated intellectual rationalism [5], those later theories fell on mostly barren ground. His disciples had gone great lengths to comprehend the complexities of refutation as a strategy

for building empirical evidence, and certainly, the idea of the null-hypothesis is stunning in its clearly structured simplicity and logic [19,20]. Thus, the researchers who had accepted and adopted the new theory of falsification as a sound strategy for testing hypotheses, had to wrap their minds around the paradoxical principle, that in order to investigate a certain scientific phenomenon, the prudent researcher constructs a hypothesis which aims to confirm the exact opposite of the expected outcome from experimentation. Rejection of such a contrasting theory does not confirm the actual theory, but it adds credibility in a culture distrusting confirmation, or rather, a culture which has elevated this concept into an unobtainable ideal.

Although these epistemological discussions certainly still bear relevance, other theories from the mind and hand of esteemed professor Popper must be examined here—notably those on alternative fundamental approaches for construction of knowledge [5]. Arguably, our educational systems are fundamentally built on the principle of filling young minds with subjectively selected facts and thus unsuited for nourishing independent thought. Alternative educational theory could investigate whether the receptiveness of human minds should instead be envisaged as heterogenous, and dissimilarly rigged cognitive machinery; and subsequently, whether diversity of mental constellations possibly necessitates parallel diversity in teaching; i.e., as individualized contact between teacher and pupil, thus suggesting a fundamentally different approach to pedagogics.

Advocating full-time one-to-one relations in primary schools is unrealistic, not least due to such practical constraints as the working capacity of teachers. However, transcending from primary school to higher education and our own domain in academia, it may be argued that the enthusiasts amongst us, those who have found rays of light in the brick wall confinement of education, and experienced the constructive bliss arising from mental interplay of minds, will acknowledge that precisely this, the inspirational and cognitive value of intimate mental interchange, most certainly should be employed as a formalized strategy to engage students. As well as in higher education, the mental bond between teacher and pupil can engage schoolchildren who have become unreceptive or intellectually starved in the conventional mass feeding logistics of information dissemination in the classroom forum.

In my own experience, such targeted teacher-approach may have immense impact on young minds alienated at the edge of common understanding; or even more importantly, on young minds that wrongly assume an inherent defect to be causal for their being situated outside of common understanding, and who have lost motivation for re-entry, simply due to lack of confidence. Managing to involve and recruit such a dormant or distant mind, interlocking with it and luring it into reflection and dialogue, is a great professional satisfaction for any devoted teacher.

Although possibly not evidently relevant at present, discussion of fundamental pedagogics is essential in understanding situations outlined in the following sections, and arguably conceivable as a profound crisis in academia. Surely, numerous academics have experienced the surprising fact that intelligence or open discussion is not always welcomed in academic circles. And, it must be highlighted as a most unfortunate aspect of mature academic communities, that although most university departments surely crave to recruit the most brilliant of creative minds, other academic brotherhoods employ a diametrically contrasting strategy: Vigorously ensuring that the newly recruited colleagues shall not become intellectually threatening to the establishment. In its most extreme form, such mismanagement is upheld by an entrenched flock of resident professors, pompously clad in wigs and walrus skin.

Discussing indoctrination and socialization in educative framework, as distilled from the layers of knowledge in Figures 2 and 3, it may be argued that, unfortunately, we mostly only see the outer layer, the contemporary surface layers of scientific heritage. Implicitly, the knowledge from predecessors which is visible and readily accessible, is the knowledge, theories and findings which have been grasped and understood, given value as interpretations, and thus brought forward as fundamental curricula for contemporary practitioners of a certain discipline: As an example, we could argue that for contemporary geneticists, biologists and molecular biochemists, knowledge of evolution is based in a brief and condensed interpretation of Darwinian theories. The writing of Jean-Baptiste Lamarck

(1744–1829) and St. George Mivart (1827–1900) may to some degree be known by specialized scholars of evolutionary theory, but the contemporary geneticists who struggle to comprehend the innermost secrets of heredity, have mostly been socialized into a certain school of theory, based on a condensed and arguably somewhat impoverished interpretation of Darwin and Mendel.

The consequences of this reduced diversity of thought are obvious, as several research efforts have marched into 'blind alleys' or have become stuck in the quagmires of dark-matter aggregations, from which they seemingly stubbornly refuse to exit. Arguably, the concept of Science Technology Studies (STS) should be paraphrased as 'Science Transformative Studies,' appropriately acknowledging the impact of reflective argumentation advanced by principal investigators; e.g., Wynne in his notable work on scientific mismanagement of a national radionuclide risk assessment program: Following the discovery of radioactive contamination of British grasslands and local food sources, the most prestigious scientific experts arrogantly employed authority and mandate, delegated from society, to create factual confusion which subsequently led to public alienation and mistrust. Paradoxically, this was through deliberated agnogenesis disguised as large-scale scientific risk-assessment investigation: Although it was clear to leading researchers and politicians at the time that the radioactive contamination originated from the local nuclear waste reprocessing industry in Sellafield, the investigation aimed at demonstrating that radioactive fallout from the Chernobyl incident had caused the contamination [21].

Other aspects pertain to researcher's focus: Analysis of contrasting research results have shown that such research can often be grouped by factors, of which author affiliation is amongst the strongest. Hence, there is a clear tendency for scientists employed in medical industries to find less adverse effects and more beneficial effects of the active ingredients manufactured by their same industry, than found in research performed by independent scientists investigating the same questions, and even using same methods. This may not come as a surprise for most professionals. However, it highlights the need for continued independent research into industry products such as new chemicals, pharmaceuticals and other inventions [3].

## 6. Perceiving the Studied Subject

Contemporary science endures several prolonged challenges from the agnotology caused by the white lacunae and dark-matter aggregations discussed earlier. Such scientific ignorance stands as an elephant in the room while researchers attempt to work around it. Many compliments can be bestowed upon pachyderms, but such a colossal piece of biomass tends to obscure the view of the surroundings—the closer we get to the elephant, the less else we see.

Arguably, the sciences originate in observation and are thus a discipline which is highly dependent on visual clues. We perceive that which we see—be it with our naked eyes, through the magnifying glass of the microscope, or through the lenses of astronomer's telescopes. Obviously, the use of echo sounders, radio-telescopes and other remote-sensing technology is all dependent on interpretation—thus the signals gained through such apparatus is conveniently transformed into depictions which can be seen.

We make models and illustrations, all in order to make the studied object visible and open it for analysis. Thus, it is appropriate to ask whether our eyesight is trustworthy and whether our models are correct depictions. Additionally, the questions should aim deeper than simply investigating whether we are able to truly see that which we study: At times we forget that complex phenomena cannot be taken out of context, and thus we conveniently convince ourselves that what we see is the full and only truth stemming from the studied subject.

However, at times such assumptions prove painfully naïve, as can be learned from the rather amusing lesson by the academic advocates of the brown trees in past centuries: John Constable (1776–1837) was an artist; a painter who achieved great fame for stunning landscapes and portraits. When Constable studied at the Royal Academy of visual arts in London, he encountered several influential figures of the arts world. Amongst those was Sir George Beaumont (1753–1827), a nobleman and art collector. At that time, student artists at the academy were copying work of predecessors to learn the craftsmanship of painting: Beaumont was very fond of the landscapes of old Italian masters,

in which all vegetation was mostly painted in somewhat dull brown hues. Beaumont argued that "a good picture, like a good fiddle, is always brown and one should always include a brown tree in every landscape." Thus, Beaumont insistently persuaded Constable to abstain from using the green colors, which the latter preferred, and follow the brown style of the Italian masters. This so provoked Constable that it almost led to a break between the two, as Constable reportedly exclaimed that he would never put such a thing as a brown tree into his landscapes [22]. Thus, Constable rebelliously painted green trees while most other students, as well as established British artists at the time, were loyal to tradition, and thus obediently and meticulously painted brown trees with brown leaves onto the brown scenery of springtime landscapes. Only much later, through a bit of scientific intervention, was it discovered that the brown trees of the Italian renaissance were not really brown: They had originally been painted with green pigments which over the course of decades and centuries had oxidized, lost their lush green luster and turned into the somewhat fecal brown nuances adored and propagated by Royal Academy instructors.

Hence, even when we manage to see the studied subject and can agree on what we see, we still may happen to arrive at fundamentally erroneous deductions. As a parallel, we could mention that when a group of astronomers recently managed to acquire an image of a black hole in April 2019, they decided to convert the registrations from numerous radio telescopes working in parallel, into an image which could be seen by the human eye. Thus, they constructed an orange halo of glowing light surrounding a circular dark center, an image which was subsequently shown on news broadcasts globally and enthusiastically presented to the public. Possibly a slight improvement on our previous illustrations of black holes, which were, well, just black holes. However, the glowing orange halo was nowhere near the complexity of radio telescope readings, which naturally are invisible to the human eye.

Thus, we must acknowledge our limitations, not only pertaining to perception of the studied subject (Figure 4), but notably also in subsequent analysis of our observations. Moreover, as if that in itself was not complex enough, we must also face the thought-provoking paradox of epistemology by which science, when analyzing itself and evaluating potential as well as limitations, only has scientific method to work by [23]. Accordingly, in order to investigate the fundamental disorder of existing scientific methods we use the best possible tools at our disposal, which obviously are exclusively found in existing scientific methods. Consequently, the only way out of this circular trajectory and academic marshland, is through rebellious innovation: We must allow upcoming generations of scientists to paint trees which are not only green, but also yellow, blue and pink, if this is what it takes to advance our stalled progress in important scientific disciplines.

In any professional discipline, language bears information and is the primary means of communication and dissemination of knowledge. Evidently, in order to uphold function, consensus must exist amongst practitioners regarding the meaning of specific words and phrases. But interestingly, even in exact academic disciplines, such as the natural sciences, the meaning of words tends to change with the passing of time. An example of this is the word "materialism," which in its contemporary interpretation designates the tangible physical matter, as opposed to the diffuse essences of the metaphysical realm. However, for scientists in the 19th century, such as the German biologist and artist Ernst Haeckel (1834–1919), materialism had a very different meaning, and from reading his books we understand that at that time, materialism was a concept which included spiritual aspects.

Thus, we may claim, that each specific time is encapsulated in its own societal and cultural matrix, a sort of contextual sphere—a *Weltbild*. And, this Weltbild is not a static figure—it evolves and gradually changes. This will be evident to anyone who has lived in a specific locality or country long enough to be fully socialized into its culture, and who then leaves for a decade or two. Upon returning, one will find that language has changed, other words are dominating the public discourse and people have changed behavior. To the locals, who have gradually co-evolved, such changes may not seem apparent.

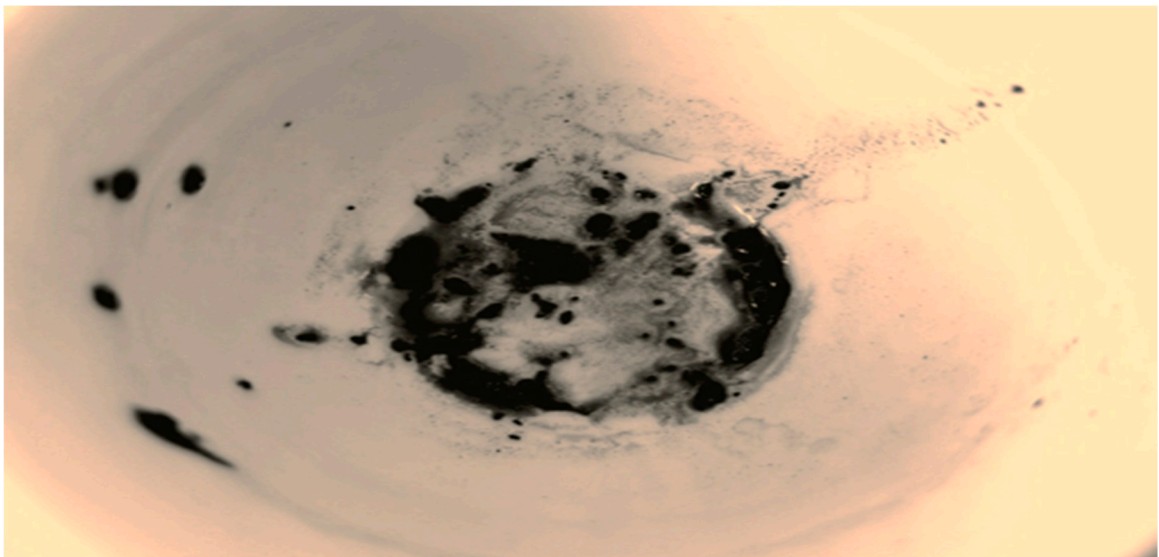

**Figure 4.** Energy dispersal in the Crab Nebula: An artist's interpretation.

Accepting such constant and gradual evolution of normality, even in sciences, not only language but also context, is imperative for understanding the writing of predecessors. Thus, we realize that our predecessors were writing in the scientific environment of their time, in which certain phrases or expressions were used differently than in the contemporary time of 2019 or in the future present of the next century. Thus, it must be suggested that retrospect shall cast judgment lightly and with caution, as we should accept that ancient texts which we are reading 20, 50 or 150 years after their composition, were written in a different Weltbild.

As a student of science was preparing to defend his thesis on risk-assessment of glyphosate-tolerant biotechnology for agriculture, a friend asked, "Why is the herbicide called Round-up, when the fields are all square?" Although such a comment may seem irrelevant, or sarcastic, or even naïve, it was served by an eager and rightful mind, and touches several interesting aspects which could be discussed: The most obvious answer to the question, would be to use the method demonstrated by the writer Antoine de Saint Exupéry (1900–1944), as he let his fictional pilot character draw a lamb by simply sketching an image of a square box and then stating, that the lamb is inside the box [24]. Arguably, a large proportion of our scientific evidence is of this nature; e.g., in genetics, as we identify various sequences of genetic code and conclude that this specific code is indicative of certain phenotypic characteristics. Thus, we acquire sufficient experimental evidence to be able to verify the existence of the box, and then we deduce that there is a lamb inside.

By similar rationale, the drawing in Figure 5 approaches an answer to the difficult question presented prior to the defense of the thesis on risk-assessment of chemical agriculture: Using toxic pesticide chemicals in order to produce food, is more or less equal to forcing a square peg into a round hole. The fundamental paradox of systematically adding toxins in the production of food has arisen slowly, with the aid of expert scientists and as a change of normality—a gradual evolution of context and Weltbild.

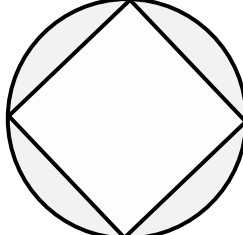

**Figure 5.** Squaring the circle: An optical illusion of discomfort.

## 7. Theoreticians and Practitioners

*"Science cannot make progress without the action of two distinct classes of thinkers: the first consisting of men of creative genius, who strike out brilliant hypotheses, and who may be spoken of as 'theorizers' in the good sense of the word; the second, of men possessed of the critical faculty, and who test, mold into shape, perfect or destroy, the hypotheses thrown out by the former class."*

—Mivart, The Essays, 1892.

Recently, the Journal of Biological Physics and Chemistry published a paper in which this author suggests a new theory on the genesis of genes: The manuscript lists and discusses published indications, and presents an interpretation that serves to construct specific hypotheses, which subsequently can be tested experimentally. Although the theory is controversial, by speculating that genes do not *per se* evolve in multicellular higher organisms but are provided through horizontal gene transfer via microorganisms, it was accepted for publication [25].

Arguably, such novel theories are not habitually welcome in science. Consequently, as scientists, only hesitantly do we present theories of which we have little certainty. Even when such theories potentially furnish explanatory value, we guard our thoughts and systematically gather evidence before publicly presenting to a wider audience. However, situations may arise when individual researchers feel compelled to disseminate theories hampered by lack of empirical evidence, realizing that such imperfect theories may inspire peers and thus lead to subsequent and less imperfect explanations.

Although preliminary or embryonic theories should be seen as constructive contributions, as personal opinions or as rational deductions stemming from a career in science, they are not always welcome. Habitually, editors and peers adhere to established truths and thus reject contrasting theory unless backed by brawny, empirically gathered evidence. Furthermore, for ideas which do get accepted and published, the exchanges of theory and countertheory can develop to be speculative or intolerantly harsh. Contrary to such conservative skepticism, even a most prestigious scientific journal may present—and thus legitimate—loosely founded theories; e.g., as seen in the interpretations of *A. afarensis* skeletal fractures, the recently discussed case of 'Lucy fell from a tree and broke her arms' [26].

The potential damage which a faulty theory or professional misjudgment can impose on the credibility of a researcher can be permanently detrimental, a fact which undoubtedly rests soberingly in the back of the mind of any scientific writer. Thus, some of us are perpetually torn between the urge to present and discuss new ideas on all sorts of exciting subjects, and the rational fear of unwillingly committing a professional hara-kiri by advancing ideas that are arguably naïve, unformed or simply incorrect.

Consequently, a cautious selection of language enshrouds many of our contributions which thus, by such hesitant approach, may become harmlessly uncontroversial. But, unfortunately, also somewhat un-interesting or even irrelevant. It must surely be prudent to bear in mind the words of professor emeritus Terje Traavik of the University of Tromsø, as he used to lecture his audiences: "If you don't have anything to say, please don't say it." Paradoxically, assessing the contemporary multitude of published scientific papers, we could argue that although many authors apparently have had nothing to say, they have said it anyway. More importantly, many ideas and theories which

could and should have been openly presented and discussed, are remaining undisclosed until either empirically vindicated or falsified [25].

Thus, it may be both speculative and immensely relevant to reflect on a few examples from the scientific realm. For the purpose of illustration, the examples could be taken by looking inward, into the innermost molecular structures of heredity and the fundamentals of epistemology, or by looking outward into the most far-reaching astronomy: Recently, stunning photos of asteroid 486958–2014 MU69 were provided by the NASA New Horizons probe [27] (Figure 6). The spectacular appearance of this 32 km long asteroid has led observers to suggest that the two lobes are formed by previously separate asteroid rocks "that orbited one another closely in a slow dance before merging" [28]. This evidence of slow fusion of such relatively large asteroids presents important indications which furnish explanatory models for understanding of the principal methods of planetary coagulation, as objects within parallel circulatory orbit in the same orbital plane and upholding similar velocities gradually experience gravitational interaction, potentially harmonizing trajectories as well as velocities and resulting in slow fusion.

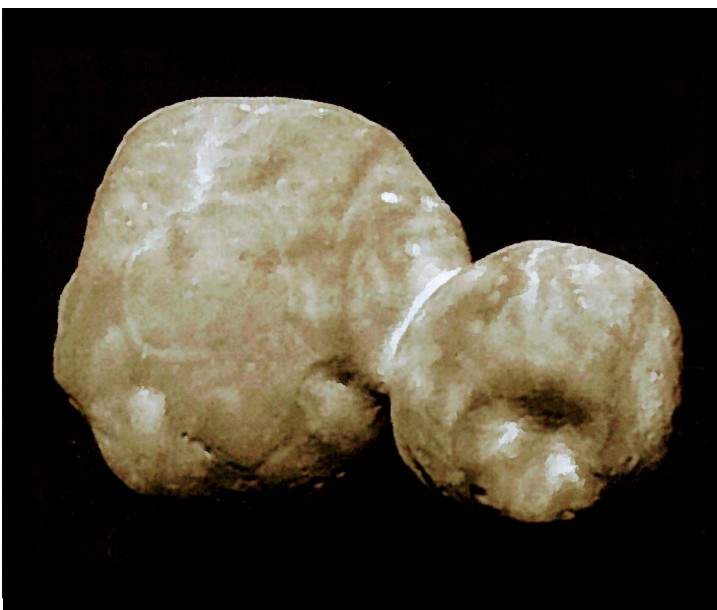

**Figure 6.** Asteroid 486958-2014 MU69 (photo courtesy of NASA).

Such hypothesis could have important explanatory value for resolving questions such as the long-standing enigma of dinosaur weight; i.e., acknowledging that an eight-ton bipedal predator, such as the *Tyrannosaurus rex*, would clearly be better suited for an environment with less gravity than contemporary Earth [29]. Furthermore, the theory of slow-fusion of large celestial objects on harmonized trajectories, in which glowing liquid cores are contained by the cool skin of outer lithospheres, could explain the geographically heterogeneous occurrence of planet Earth's fossilized communities; e.g., found in the environments of the Burgess shale and other such conserved mosaic patches, as tiles of frozen diversity on the contemporary planetary surface. Although astronomers at some research facility somewhere may at present be engaged in elaborate observations and calculations in order to further validate such a hypothesis, other scientists will not know, because of the aforementioned culture of keeping new ideas 'cloaked until empirically vindicated' [25].

Evidently, presenting a critique of scientific culture and scientific method is not by itself constructive, unless followed by practical suggestions for remediation and improvement. A previous contribution lamented on the observation that although there are at present a wide plethora of scientific journals within specific subjects, these journals arguably mostly all adhere to a common and rather rigid form of presentation [26]: A contemporary scientific paper is expected to consist of a clear and short

title, information on authors names and their affiliations, an abstract, an introduction, a chapter detailing materials and methods, and a discussion followed by a concise conclusion and a list of references. Although style of referencing may vary slightly, the general layout and structure of scientific publications adhere rather dogmatically to this exclusive norm. Undoubtedly, such structure aids most of us by providing a clear-cut template for our research tasks, easing our efforts when planning, executing and reporting scientific work. However, this structure has become so dominant in scientific publication, that many editors and reviewers will *a priori* reject contributions which do not follow such standards.

Obviously, any scientist must tread carefully when voicing critique or advancing revolutionary suggestions, such as the use of footnotes in natural sciences [26]. Thus, short of advocating rebellion against the established system norms of publication, the author wishes for a small change relating to the present normality: Scientific papers could have an appendix or post-script; e.g., bearing the heading "Perspectives" or "Implications," which would allow the author(s) to present a few open reflections pertaining to the work. Such reflections which have arisen during a specific scientific investigation and can be seen as stemming from it. Thus, any given research into a specific subject would still present its essential findings according to established methods, but also have an optional dedicated section which would allow for scientific reflections reaching further than the narrow focus of the studied subject. At present, such unfounded reflections are mostly perceived as speculative and are habitually rejected by editors and reviewers, even when relevant as well as important.

Gaius Plinius Secundus (AD 23-79) was a Roman naturalist, writer and politician. Acknowledging the impact of his encyclopedia on natural history, *Naturalis Historia*, Plinius was amongst the first scientists to shape the foundation of our understanding of the natural world: Arguably, the 12 encyclopedic volumes of his work are at the center of our sphere of scientific knowledge. Amongst many other phenomena, Plinius wrote of *Ultima Thule* as an island far north of the British Isles; an island where the sun remains in the sky and does not set. Off course, to his contemporaries and successors, the speculative idea of a place where the sun constantly hovers above the horizon for periods of several weeks or even moths, must have seemed absurd and against all established logic. Later, the concept of 'Ultima Thule' has been used by other authors to describe localities or objects at the outer perimeter of our cognitive understanding and physical reach. Befittingly, when the strange twin-asteroid 486958-2014 MU69 was discovered in the Kuiper belt at the extreme periphery of our solar system, it was given the name *Ultima Thule*.

When Dr. Donald Williamson (1922–2016) proposed a controversial theory of species hybridization, as explanation for anomalies observed in many decades of studying echinoderm embryogenesis and larval metamorphosis of other invertebrate species [30,31], his ideas were rejected by fellow scientists, until finally a brave editor dared by publishing his work [32]. When Dr. Paul Kammerer (1880–1926) presented his theories of phenotypic plasticity and experimental revival of inherent (dormant) physiology in developing vertebrates and invertebrates [33], he arguably was met with not only skepticism but also systematic defamation [34]. When Dr. Lynn Margulis (1938–2011) attempted reviving the theories of endosymbiosis presented half a century before by Dr. Kozo-Polyansky (1890–1957) [35], she was ridiculed, and her manuscripts suffered numerous rejections.

When Dr. Stephanie Seneff recently submitted a fifth detailed manuscript presenting indications of a hitherto ignored and potentially detrimental mode of toxicity of the globally dominant herbicide glyphosate, which is a synthetic analogue of the vitally necessary amino acid glycine, she envisaged that many journal editors would *a priori* reject her work as *speculative*—arguably because she as a scientist has been labeled as an activist (personal communication).

Thus, the history of natural sciences bears numerous examples of suppressed or misunderstood contributions, which either have been long forgotten, or, as in the case of some of the scientists mentioned above, have been vindicated and received appropriate attention only due to insistent proponents who defended the work in scientific circles and helped it gain broad acceptance.

Regarding the work of Dr. Lynn Margulis, who argued that mitochondria and chloroplasts in eukaryote cells had originated from adapted as well as adopted prokaryote endosymbionts, her ideas eventually gained acceptance and acclamation, but she had to endure harsh criticism from fellow scientists who would not accept her argumentation [36]. Reportedly, upon reviewing a manuscript submitted by Dr. Margulis, one of those fellow scientists suggested that Margulis should leave science and find something else to do.

Naturally, it takes extraordinary stamina and resolve to keep defending and presenting controversial or unwanted theories when facing opposition, which can be fierce and based in a principal conviction that debunks alternative explanation, simply because such explanations go against established truths or transcend the limitations of the recipient's personal mind [37]. However, sometimes the foundations of established truths are embedded in the dark-matter aggregations visualized in Figure 3, and new light must be allowed to shine on them in order for science to progress.

Unfortunately, the evolutionary path of natural sciences is paved with headstones of scientists who were misunderstood or ignored in their time and age. Gratefully, although the heritage of published evidence is immense, much of this heritage is increasingly accessible, as old volumes are digitized into databases and open repositories such as the *Gutenberg Project* and the *Biodiversity Heritage Library*. Thus, any meticulously engaged researcher can access and interpret original sources and disregard such established interpretations, which may or may not serve justice to the original writing.

Several factors still stand as major challenges to this dissemination of heritage, notably the priorities demonstrated by experts who select work for digitalization, and importantly, the hindrances caused by language: Contemporary sciences are almost exclusively conducted in the English language, albeit previous centuries have furnished contributions in Czech, Danish, German, French, Latin, Russian, Spanish and Italian, just to name a few, and remain centered in a European perspective. Obviously, ancient Greek texts by naturalist philosophers are a known foundation for European sciences, but the European knowledge of theories of Asian medicine and other texts in, e.g., Sanskrit or Mandarin languages, has been unfortunately limited. Regrettably, the global evolution of natural sciences around the common language has wrought socialization and homogenization onto scientific communities. Thus, it may be claimed that although more than seven million contemporary scientists world-wide presently produce more than two million papers annually [38], the global diversity of reflective thought is unfortunately limited and unnecessarily low: It has never been poorer.

## 8. Concluding Remarks

Writing these reflections from personal experiences harvested through three decades in academic employment, and from numerous encounters with a wide array of professional topics and research questions in science; the ambition was to gather examples of a selected few enigmatic unanswered questions and ethical challenges from my own practice, and contrast these with principal defects in natural sciences; as interpreted from not only contemporary findings, but also the heritage of published literature.

Literacy at the receiving end is surely still a prerequisite for successful writing, and, although the simple task of structuring reflections into sentences in itself can be quite fulfilling, it cannot be our main motivation [39]. Writers of science, poetry, fiction or journalism all adhere to the same fundamental rule of text production: We write for the reader. In science, the reader is mostly a fellow scientist, but that does not imply that we have to communicate as if that reader is *just* a scientist—because, the scientist is also simply and merely *human*, with the strength, weakness, courage, fear, ambition, laziness, altruism, greed, indifference and passion inherent in any human soul.

Thus, attempting to write a critique on such ethical issues, any author must face a classical and unpleasant question: Who amongst us may feel justified to throw the first stone? Who amongst us has never done wrong? Any scientist present at the aforementioned receiving end at this very moment in space and time might feel intimidated by the style and language employed here, which admittedly wrenches the writing onto a path leading into epistemology, agnotology and philosophy,



and possibly even more obscure territories where practitioners of natural sciences only too seldom venture. Nevertheless, the main message is simple—there are severe challenges to our continued existence on this planet, and only reliable research may provide the solutions and recommendations which societies and responsible politicians need, in order to make decisions and shape a policy which decelerates the decline: Mankind urgently needs to restructure the mismanagement of nature and reduce our destructive consumption of natural resources. Thus, the present commentary aspires to raise fundamental ethical questions related to the present research efforts, as outlined in the large national and international strategies, such as assessments of climate change and the struggle against cancer and intoxication of environment.

Initially, the working title was, "Academic Arrogance Essentially Conserves Dogma in Natural Sciences." Such words may furnish some explanatory value but were deemed too blunt and contrary to necessary diplomacy, as the ambition of the writing is to initiate academic dialogue. Thus, a more pompous but less intimidating title was thought up: "Contemporary natural sciences: Challenges and possibilities." However, the enormity of a general and exhaustive analysis of the challenges to our profession, coupled with the intention to provide solutions, was a bit too ambitious for a writer who is mainly a laboratory researcher, and a trained theoretician of neither epistemology nor philosophy. However, as the author is a signatory of the aforementioned initiative: "World Scientists' Warning to Humanity: A Second Notice" [1], the choice of title was meant to open reflections on the responsibility of our profession.

Although every dedicated and honest professional amongst academics ought to perpetually reflect upon fundamental issues, such as whether his or her work is significant, important, representative and truthful, to such a degree as to deserve dissemination amongst peers and public, it unfortunately seems that such hesitation can be interpreted as not only counter-productive in the academic world at present, but fundamentally obstructing own career. The principle defined as 'publish or perish' surely explains the present situation, in which we are inundated in publications, of which a large part, objectively, are only adding to the elephant in the room and obscuring the view of the more important issues.

**Funding:** This research is supported by the Norwegian state via Havforskningsinstituttet. Initially funded in 2008 by the Research Council of Norway through NFR Project 184107/S30 LAND: A new model approach to assess genetically modified plants: their ecotoxicity and potential interactions with environmental pollutants.

**Acknowledgments:** To Brian—as one of several ideas, the theory on lacunae and dark matter aggregations arose during our inspirational discussions in June 2015 and has subsequently been brewing. The writing was structured during a few concentrated days in March 2019, offline and isolated from disturbances at the Polar Cabin at Vaagnes, a retreat in the mountains of northern Norway. The author is grateful to Michal and Jana for their hospitality.

**Conflicts of Interest:** The author declares no conflict of interest.

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
