# Peer review of "The Scientist: Creator and Destroyer—“Scientists’ Warning to Humanity” Is a Wake-Up Call for Researchers"

_challenges, doi:10.3390/challe10020033_

Round 1

Reviewer 1 Report

The title of the article is a little brief and has a commercial character but does not inform about the main question wich is the responsibility of scientist. It will be good to add a subtitle which will inform about the main question of the article.

Reviewer 2 Report

This is well written and thought provoking. The author challenges us to monitor ourselves, our socialization, and the potential for blinders to new ideas. Significant questions are asked.

Here is a short expansion of my thoughts. I believe that the questions raised are thought provoking and significant for the development of new knowledge.  I believe in the value of questioning ourselves and expanding the methods and theories we are willing to explore.

This was a well written manuscript that raises questions about the ways educators, researchers, and scholars limit the ability of our peers to question our assumptions and methods. The author is thorough exploring multiple strands of knowledge and philosophy in framing the limits and possibilities. In challenging us to question ourselves, a path may be opened for considering the potential for us to expand our knowledge by taking a risk in posing alternative models or methods.

Reviewer 3 Report

This paper contains many interesting ideas, and that main thesis it puts forward is a plausible and important one, namely that scientists need to become more sensitive to and take more responsibility for the way their output is understood and misunderstood by the greater public, and to put more effort into shaping how it is used. But I was unable to follow the main argumentative line in support of that thesis through the paper. A roadmap in the introduction, that outlines for the reader how the paper is going to proceed, what will be discussed in each section, and how the different sections fit together to support the main thesis, will be very useful. As would a concluding section that once again concisely stated the main thesis and argument that the paper has developed in support of it.

So: my main concern has to do with argumentative structure. As it is, the overall logic of the paper continues to elude, and there were what seemed to be several lacunae in the reasoning that I took to be in play connecting the more local dots into a larger argument. Displaying that logic and making explicit the way the pieces fit together will be the most important thing to do to improve this paper going forward. Turning up the focus on a single theory or theorist, to serve as a stalking horse or foil that the author is contrasting his/her own view with may provide a way to do this, within a familiar dialectic structure. This may necessitate that some of the material be dropped, in order to make a more focused and plausible case for the main thesis.

A second, related concern was that the author was trying to do too much in too little space, and it was diluting from the force of the argument. Some of the discussions—on the sociology of science, for example—would be enriched by tapping into the more current literature.
